# The Role of a High-Fat, High-Fructose Diet on Letrozole-Induced Polycystic Ovarian Syndrome in Prepubertal Mice

**DOI:** 10.3390/nu14122478

**Published:** 2022-06-15

**Authors:** Joanna Maria Pieczyńska, Ewa Pruszyńska-Oszmałek, Paweł Antoni Kołodziejski, Anna Łukomska, Joanna Bajerska

**Affiliations:** 1Department of Human Nutrition and Dietetics, Poznań University of Life Sciences, 60-637 Poznań, Poland; joanna.pieczynska@up.poznan.pl; 2Department of Animal Physiology, Biochemistry and Biostructure, Poznań University of Life Sciences, 60-637 Poznań, Poland; ewa.pruszynska@up.poznan.pl (E.P.-O.); pawel.kolodziejski@up.poznan.pl (P.A.K.); 3Department of Preclinical Sciences and Infectious Diseases, Poznań University of Life Sciences, 60-637 Poznań, Poland; anna.lukomska@up.poznan.pl

**Keywords:** polycystic ovary syndrome, pre-pubertal mice, high-fat and high-fructose diet, metabolic disorders, endocrine disorders

## Abstract

This study aims to investigate the effects of a high-fat, high-fructose (HF/HFr) diet on metabolic/endocrine dysregulations associated with letrozole (LET)-induced Polycystic Ovarian Syndrome (PCOS) in prepubertal female mice. Thirty-two prepubertal C57BL/6 mice were randomly divided into four groups of eight and implanted with LET or a placebo, with simultaneous administration of an HF/HFr/standard diet for five weeks. After sacrifice, the liver and blood were collected for selected biochemical analyses. The ovaries were taken for histopathological examination. The LET+HF/HFr group gained significantly more weight than the LET-treated mice. Both the LET+HF/HFr and the placebo-treated mice on the HF/HFr diet developed polycystic ovaries. Moreover the LET+HF/HFr group had significantly elevated testosterone levels, worsened lipid profile and indices of insulin sensitivity. In turn, the HF/HFr diet alone led to similar changes in the LET-treated group, except for the indices of insulin sensitivity. Hepatic steatosis also occurred in both HF/HFr groups. The LET-treated group did not develop endocrine or metabolic abnormalities, but polycystic ovaries were seen. Since the HF/HFr diet can cause substantial metabolic and reproductive dysregulation in both LET-treated and placebo mice, food items rich in simple sugar—particularly fructose—and saturated fat, which have the potential to lead to PCOS progression, should be eliminated from the diet of young females.

## 1. Introduction

Polycystic ovary syndrome (PCOS) is one of the most common hormonal disorders among women of reproductive age. It significantly impairs their fertility and increases the risk of obesity, type 2 diabetes, hyperlipidemia, and cardiovascular disease [1]. Although the exact cause of PCOS is unknown, recent reviews of the PCOS research have found that genetic susceptibility is associated with PCOS and that environmental factors—such as endocrine disruptors and poor diet—are likely to play an important role in the expression of those genetic traits [2]. PCOS often manifests in the early reproductive years; puberty has been suggested as a critical developmental time period for the development and pathology of PCOS [3]. One factor commonly believed to be a risk factor for the development of adolescent PCOS is excess body weight [4]. With normal body weight, the level of total testosterone physiologically increases and the concentration of sex hormone binding globulin (SHBG) decreases, leading to an increase in the concentration of free testosterone; in obese girls, these changes are much more pronounced, leading to hyperandrogenemia [5]. This occurs because excessively developed adipose tissue is unable to properly secrete leptin and adiponectin, which are responsible for regulating androgen concentrations [6]. Moreover, excessive body weight is often associated with the development of insulin resistance (IR), where higher insulin concentrations stimulate the ovaries to secrete more androgens [7].

At present, young people are increasingly subjected to significant exposure to diets rich in saturated fat, sugar, and fructose in particular [8]. Soft drinks, energy drinks, fruit juices and nectars, and more generally, free sugars in liquid form represent the main source of fructose consumed by today’s young generation, especially obese individuals. Of course, fructose is also contained in fruit, but in minimal quantities compared to the weight of the fruit itself [9]. This type of dietary environment has undoubtedly contributed to the alarming increased prevalence of childhood obesity [10]. The results of the survey by Pathak and Nichter imply that the childhood exposure of girls to a diet high in saturated fat and simple sugars may also cause hormone disturbances during the prepubescent reproductive maturation period, leading to lifelong ovarian dysfunction and the progression of PCOS [11].

Many animal models have been developed with the aim of coming to a fuller understanding of the potential mechanisms underlying PCOS. Prepubertal exposure to the aromatase inhibitor letrozole (LET) has been used for this purpose [12]. Although there are inconclusive data on metabolic disorders resulting from exposure to LET [13,14,15], polycystic changes in ovaries have occurred and disturbed endocrine parameters have been observed; these include elevated levels of testosterone and luteinizing hormone and reduced levels of estradiol, progesterone, and follicle-stimulating hormone [16,17]. The results of earlier studies indicate that the addition of high-fat diet to the LET model affords good metabolic aberrations, along with ovarian cysts [1,18,19]. However, all these studies were conducted among adult female rodents. Pilot studies employing chronic, mild elevation of androgen in prepubertal rhesus macaques maintained through young adulthood demonstrated that these female monkeys developed metabolic and ovarian dysfunction when they were exposed to a high-fat, calorie-dense diet [20,21]. These findings revealed for the first time that diet can directly modulate the reproductive and metabolic symptoms associated with hyperandrogenemia. A similar effect is caused by high-fat and high-sugar (HF/HS) diets administered in the prepubertal period. However, it has been suggested that polycystic ovaries and hyperandrogenism develop secondarily to the IR induced by the tested diet [22].

Although the increased consumption of highly processed food, rich in simple sugar particularly fructose and saturated fat, has been associated with obesity and metabolic disorders in young people, the effects of this diet on the symptoms of PCOS around the time of puberty are not clear. This study aims to investigate the effects of a HF/HFr diet on metabolic/endocrine dysregulations associated with letrozole-induced PCOS in prepubertal female mice.

## 2. Materials and Methods

### 2.1. Experimental Animals and Treatment

Every effort was made to minimize both the number of animals used and their suffering. We calculated the sample size using G*Power software (RRID:SCR_013726); the sample size of the mice was also determined in accordance with Zheng et al. [23]. The effect size was calculated to be 2.05 on the basis of the differences in HOMA-IR between the control HFD (high fat diet) group and the PCOS+HFD group. With an alpha value of 0.05, a sample size of eight mice per group would yield a power of 0.95.

Thirty-two (32) prepubertal C57BL/6 mice (average body weight 13.5 g) with an age of three weeks were involved in the experiment. The animals were purchased from the Mossakowski Institute of Experimental and Clinical Medicine, Polish Academy of Sciences, Warsaw, Poland, and were housed in the vivarium at the Department of Physiology, Biochemistry and Animal Biostructure, part of the Faculty of Veterinary Medicine and Animal Sciences at Poznań University of Life Sciences. They were allowed to adapt to the laboratory environment for ten days. All animals were housed in standard polycarbonate cages and maintained in a controlled environment, with a temperature of 21 ± 1 °C, humidity of 55–65%, and a twelve-hour light–dark cycle. After acclimatization, at four weeks of age, the mice were randomly assigned to four groups: (1) mice receiving placebo pellet fed a standard diet (*n* = 8); (2) mice receiving placebo pellet fed the HF/HFr diet (*n* = 8); (3) mice receiving LET pellet fed a standard diet (*n* = 8); and (4) mice receiving LET pellet fed the HF/HFr diet (*n* = 8). Subcutaneous implantation of continuous release letrozole (3 mg, 50 μg/day) or the placebo pellet was performed to induce PCOS or form a control group, respectively. Letrozole was purchased from Innovative Research of America.

Such young animals were used because PCOS should be induced in the prepubertal period [13]. Two groups of mice were fed with a standard laboratory diet (3.8 kcal/g, energy supply ratio: protein 18%, carbohydrate 66%, fat 16%). In turn, the other two groups were fed the HF/HFr diet (4.7 kcal/g, energy supply ratio: protein 17%, carbohydrate 37.5% (mainly fructose), fat 45.5%). The experimental diets were bought from Morawski Animal Feed (Kcynia, Poland).

The animals had unlimited access to water and food throughout the experimental period. Once a week, the animals were weighed with a Sartorius MSE2202S-100-D0 (Germany) precision balance. During the fourth week of the experiment, dietary consumption was assessed by randomly selecting four mice from each group and placing them in a semi-metabolic cage for 3 days. For this purpose, the diet provided and the diet that remained uneaten were weighted, and the difference was calculated to give the weight consumed. The study was approved by the Local Ethical Commission under permission No. 51/2021 and was performed in line with the ARRIVE 2.0 guidelines for animal research [24].

### 2.2. Sample Collection

The animals were sacrificed after five weeks of the experiment by decapitation. Blood was collected into nonheparinized sample bottles. The blood was centrifuged (3500× *g*, 15 min, 4 °C) to obtain serum samples, which were subsequently kept frozen at −80 °C until needed for biochemical assays. The liver was collected and frozen in liquid nitrogen.

Immediately after the last blood draw, ovary samples were rapidly removed from the animals. Ovaries from each mouse were fixed in 10% formalin (formaldehyde in saline). The ovaries were stored at 4 °C in 50 mL of 20% sucrose in PBS for 24 h before sectioning. Ovary sections of 4–5 μm thickness were obtained using a cryostat (CM1860 Ag Protect; Leica Biosystems, Warsaw, Poland) and collected on microscopic slides (Menzel-Glaser, SuperFrost Ultra Plus, Thermo Scientific, Budapest, Hungary). Subsequently, the sections were stained with hematoxylin-eosin (Sigma-Aldrich, Madrid, Spain) in line with the standard histological procedures; they were cover-slipped with DPX Mountant for Histology (Sigma-Aldrich, Madrid, Spain). Slides were examined under a light microscope (Leica, DM500, Leica Biosystems) and analyzed with LAS 4.9 software (Leica Biosystems). The ovarian preparations were analyzed in terms of the number of follicles and their diameters, the corpus luteum, and the thickness of the theca layer and the follicular wall.

### 2.3. Serum Biochemical Analysis

Serum glucose (GLU), triglycerides (TG), total cholesterol (TC), HDL-C, LDL-C, C-reactive protein (CRP), and total antioxidant capacity (TAC) were measured using commercially available colorimetric and enzymatic assays from Pointe Scientific (Lincoln Park, MI, USA). The concentration of non-esterified fatty acids (NEFA) was determined using an enzymatic test from Wako (Oxoid, Dardilly, France). Concentrations of insulin in blood serum were measured using an immunoassay (ELISA) kit obtained from Sunlong Biotech (Hangzhou, Zhejiang, China). The level of testosterone was analyzed using an immunoassay (ELISA) kit from LDN (Nordhorn, Germany). Liver cholesterol and triglycerides were analyzed after lipid extraction using a cholesterol and triglycerides kit (Pointe Scientific, Lincoln Park, MI, USA), as described by Folch et al. [25]. The total antioxidant capacity was measured using the TCA method with a TBARS Assay Kit. (Cayman Chemical, Ann Arbor, MI, USA). The optical density of the samples was measured using a Synergy 2 microplate reader (Biotek, Winooski, VT, USA).

### 2.4. Calculation of the HOMA-IR, HOMA-β, and QUICKI Indices

Insulin resistance and β-cell function were evaluated using the Homeostasis Model Assessment Method. The HOMA-IR was calculated using the following formula:HOMA − IR = fasting glucose [mmol/L] × fasting insulin [µIU/mL]/22.5

HOMA-β was calculated using the following equation: HOMA − β = FI × 20/(FG—3.5), where FI is fasting insulin (in µU/mL) and FG is fasting glucose (in mmol/L).

The quantitative insulin sensitivity check index (QUICKI) was calculated using the following formula:QUICKI = 1 log (fasting glucose [mg/dL]) + log (fasting insulin [µIU/mL])

Non-HDL was calculated using the following formula: Non-HDL = total cholesterol (mg/dL) − HDL (mg/dL).

### 2.5. Statistical Analysis

The results were statistically evaluated using Statistica 13.3.0 (TIBCO Software, Palo Alto, CA, USA; 2017). The results are presented in the tables and figures as arithmetic means ± standard deviations (SD), and the data in some figures are presented as medians with boxes and whiskers representing the interquartile range and the 5th–95th percentiles (GraphPad Prism 9.3.1. (471), GraphPad Software, San Diego, CA, USA). One-way analysis of variance (ANOVA) was used to compare the mean values of variables among the groups. Tukey’s post hoc test was used to identify the significance of pairwise comparison of mean values among the groups. Values with different letters (a, b) show statistically significant differences (*p* < 0.05, a < b).

## 3. Results

There were no significant differences in body weight at the beginning of the experiment (placebo-treated mice: 14.6 ± 1.2 g; placebo-treated mice on HF/HFr diet: 15.0 ± 1.5 g; LET-treated mice: 14.7 ± 1.5 g; and LET-treated mice on HF/HFr diet 14.5 ± 0.6 g; data not shown).

Both the placebo and the LET-treated group of mice on the HF/HFr diet showed significantly higher diet consumption than the corresponding controls (*p* < 0.05). Moreover, the LET+HF/HFr group ate significantly more than mice receiving the placebo (*p* < 0.05). On the other hand, placebo-treated mice on the HF/HFr diet ate significantly more (*p* < 0.05) than LET-treated mice (Figure 1A). After 35 days of the experiment, mice from the LET+HF/HFr group had gained significantly more weight (by a factor of 1.3; *p* < 0.05) than the related control. Moreover, this group of mice showed a significantly higher weight gain than placebo-treated mice either on the control diet (*p* < 0.05) or on the HF/HFr diet (*p* < 0.05; Figure 1B).

Both placebo- and LET-treated mice on the HF/HFr diet showed significantly (*p* < 0.05) higher cholesterol concentrations than the corresponding controls. Moreover, the latter group had a significantly (*p* < 0.05) higher cholesterol concentration than the placebo-treated mice. In turn, the placebo-treated mice on the HF/HFr diet had substantially higher cholesterol levels than the LET-treated mice (*p* < 0.05; Table 1). Exactly the same dependencies were observed for the concentrations of HDL cholesterol, non-HDL cholesterol, and LDL cholesterol. TG concentration and TC/HDL ratio fluctuated at the same level in all four groups, and no significant differences were observed here. Substantially higher levels of plasma NEFA were seen in the HF/HFr placebo-treated mice than in the related controls (*p* < 0.05). This parameter in the placebo-treated mice on the HF/HFr diet was also considerably higher (*p* < 0.05) than in the LET-treated group and even than in the LET+HF/HFr group. HF/HFr feeding also disturbed the hepatic lipid metabolism of the experimental mice. The hepatic concentrations of both TC and TG increased remarkably compared to the control (*p* < 0.05) after the placebo mice had fed on the HF/HFr diet. The hepatic concentrations of TG in this group were also significantly higher than in the LET-treated group (*p* < 0.05). Additionally, the hepatic concentrations of TG were also significantly higher (*p* < 0.05) in the LET+HF/HFr group than in the related control. In this group of mice, the hepatic concentrations of both TC and TG were also significantly higher than in placebo mice fed a standard diet (Figure 2A,B). Plasma glucose levels were significantly higher (*p* < 0.05) by a factor of 1.2 in the placebo-treated mice on the HF/HFr diet than in the LET-treated mice. The LET+HF/HFr group of mice had significantly higher values (*p* < 0.05) of the HOMA-IR and QUICKI indices than the related controls. No significant differences in HOMA-β and CRP levels were observed between groups, while the placebo-treated mice fed the standard diet had a significantly higher TAC, by a factor or 4.9, than the LET+HF/HFr group.

Serum testosterone concentration was robustly higher by a factor of 1.6 in the LET+HF/HFr group of mice (*p* < 0.05, Figure 3) compared to the control. Moreover, testosterone levels in the placebo mice on the HF/HFr diet were significantly higher (*p* < 0.05) than in the LET-treated group. In the LET-treated group, the LET+ HF/HFr group, and the placebo mice on the HF/HFr diet, an increased number of corpus luteum were observed compared to the placebo-treated mice on the standard diet. Moreover, the placebo mice on the HF/HFr diet had the largest number of ovarian follicles of the four groups. This group also had the thickest granulosa layer in the follicle, while in both the LET-treated and the LED+HF/HFr groups, this layer’s thickness was visibly reduced. Furthermore, the LET+HF/HFr group had the greatest follicle diameter and the thickest theca folliculi. Follicular atresia and the presence of cysts were observed not only in both the LET-treated groups but also in the group of placebo-treated mice on the HF/HFr diet. Interestingly, the changes in ovarian morphology in the LET-treated mice were milder than in the LET+HF/HFr group, at least partly due to the smaller number of cysts and lesser degradation of the granular cell layer (Figure 4, Figure 5, Figure 6, Figure 7, Figure 8 and Figure 9).

## 4. Discussion

Since PCOS often manifests in the early reproductive years, puberty is considered to be a critical time period for the development of PCOS. Indeed, our study demonstrated that five weeks of HF/HFr feeding initiated at the prepubertal age provoked some reproductive and metabolic features of PCOS in LET-treated female mice. More specifically, in the LET+HF/HFr group of mice, elevated testosterone levels and morphological changes in the ovaries were seen, suggesting PCOS. Indeed as was shown previously, elevated serum testosterone levels are related to endocrine imbalances and contribute to PCOS symptoms such as ovarian dysfunction and irregular ovarian or estrous cycles [26]. The LET+HF/HFr group of mice also had a worsened lipid profile (TC, LDL-C, HDL-C and non-HDL, except of TG) and insulin sensitivity indices (HOMA-IR and QUICKI). Hepatic steatosis also occurred in this group of mice. Our findings are in line with those of previous studies showing that the use of letrozole with a high-fat diet may induce or worsen the symptoms of PCOS [1,25]. More specifically, Xu et al. noted that twelve weeks of administration of LET with a high-fat diet in female Sprague–Dawley rats induced anovulatory cycles and polycystic ovary morphology, body weight gain, elevated testosterone levels, abnormal glucose and lipid metabolism, as well as insulin resistance [27]. Begum et al. indicated that twelve weeks of administration of the LET+HF diet in Wistar female rats induced additional glucose intolerance [1]. It should, however, be highlighted that all PCOS symptoms seen in our study developed as early as week five of the experiment when LET+HF/HFr was administrated; moreover, hepatic steatosis also occurred. Worsened insulin sensitivity indices, as observed in the LET-treated mice on the HF/HFr diet, seem to be associated with the excess body weight of mice. Indeed, obesity is associated with inflammation and the generation of reactive oxygen species that have a potent role in inducing insulin resistance [15]. In line with this, in our LET-treated mice on the HF/HFr diet, we observed substantial decreases in total antioxidant capacity (TAC) compared to placebo-treated mice on a standard diet. This indicates that the occurrence of PCOS is associated with oxidative stress in PCOS women, which may even contribute to the pathogenesis of this disorder [28]. Agreeing with this, a case-control study showed statistically significant decreases in the TAC levels of women with PCOS as compared to the control group [29].

Interestingly, it was seen in our study that LET itself did not lead to the development of any endocrine or metabolic abnormalities in experimental mice, but polycystic ovaries were observed. In contrast, Arroyo et al. and Skarra et al. demonstrated that five weeks of LET treatment resulted in the hallmarks of PCOS, including elevated testosterone and luteinizing hormone (LH) levels, acyclicity, and the appearance of cystic ovarian follicles [13,30]. However, despite the hormonal variations shown in different animal studies, letrozole in general manifests good reproducibility for PCOS-like features in rodents [15] and is believed to cause the lean reproductive phenotype of PCOS [31].

The HF/HFr diet itself may also lead to some features of PCOS. As was observed in our study, five weeks of exposure to the HF/HFr diet significantly elevated serum testosterone of the female mice and also disturbed some lipid parameters (TC, LDL-C, HDL-C and non-HDL, except of TG). Elevated levels of glucose and polycystic changes in ovaries were also observed. However, worsened insulin sensitivity was not observed in this model. Roberts et al. indicated that a high-fat, high-sugar diet given for eleven weeks led to hyperinsulinemia but not to hyperandrogenemia in experimental rats [22]. However, elevated testosterone levels in that study were predictive of a high number of ovarian cysts [22]. In our study, the elevated testosterone levels seen in a group of mice on an HF/HFr diet seem to be the effect of higher levels (though not statistically significant levels) of insulin. Indeed, insulin acts directly through its own receptor in PCO theca cells to increase androgen production [32]. Interestingly, metabolic disturbances in the placebo-treated mice fed the HF/HFr diet were seen even when the body weight of those mice did not increase significantly. It was also surprising that, despite the equally high consumption of the HF/HFr diet, only the LET-treated mice gained significantly more weight, while the body weight of the placebo mice did not differ from that of the other groups. Similar results were obtained by Patel and Shah, but this was associated with a reduction in food intake, which was not observed in our experiment [33]. Huang et al. explained that female rodents are relatively resistant to hyperphagia and weight gain in response to a high-fat diet, in part due to the effects of estrogen, which suppress food intake and increase energy expenditure [34].

Since increased prevalence of NAFLD has been reported in women with PCOS [35], we also assessed the hepatic accumulation of TC and TG in our experimental mice. More specifically, the hepatic accumulation of both TC and TG increased remarkably after the placebo-treated mice were fed the HF/HFr diet, as compared with control. In LET-treated mice on the HF/HFr diet, only the TG level was significantly higher than that of the related control. The accumulation of excess triglycerides in hepatocytes is generally the result of the increased delivery of non-esterified fatty acids (NEFAs), increased synthesis of NEFAs, impaired intracellular catabolism of NEFAs, impaired secretion as triglyceride, or a combination of these abnormalities [36]. In our study, only higher levels of the plasma NEFA in the mice fed a HF/HFr diet compared to placebo-treated mice was seen. Interestingly, despite the visible NAFLD, we did not observe significant differences in serum triglyceride concentration, as hypertriglyceridemia develops secondarily to hepatic steatosis [37] and the period of 5 weeks was likely insufficient for its full appearance [38].

## 5. Conclusions

Our findings reveal for the first time that HF/HFr feeding given around puberty may directly stimulate reproductive and metabolic symptoms, not only in LET-treated mice but also in placebo-treated mice. These findings indicate that a diet that is highly processed, high in simple sugars (particularly fructose), and high in saturated fats may, if eaten every day, have a great impact on PCOS progression in young females. Food products that are rich in these ingredients should therefore be eliminated from the diets of women with PCOS, especially younger women. Furthermore, the combination of the HF/HFr diet with LET causes visible metabolic disorders, comparable to those found in women with PCOS. Moreover, hepatic steatosis also occurred. Thus, this animal model can be used to the test various options for PCOS treatment. In turn, the model based on letrozole alone was not sufficient to induce the above-mentioned disorders, although it caused visible changes in the morphological structure of the ovaries.

## Figures and Tables

**Figure 1 nutrients-14-02478-f001:**
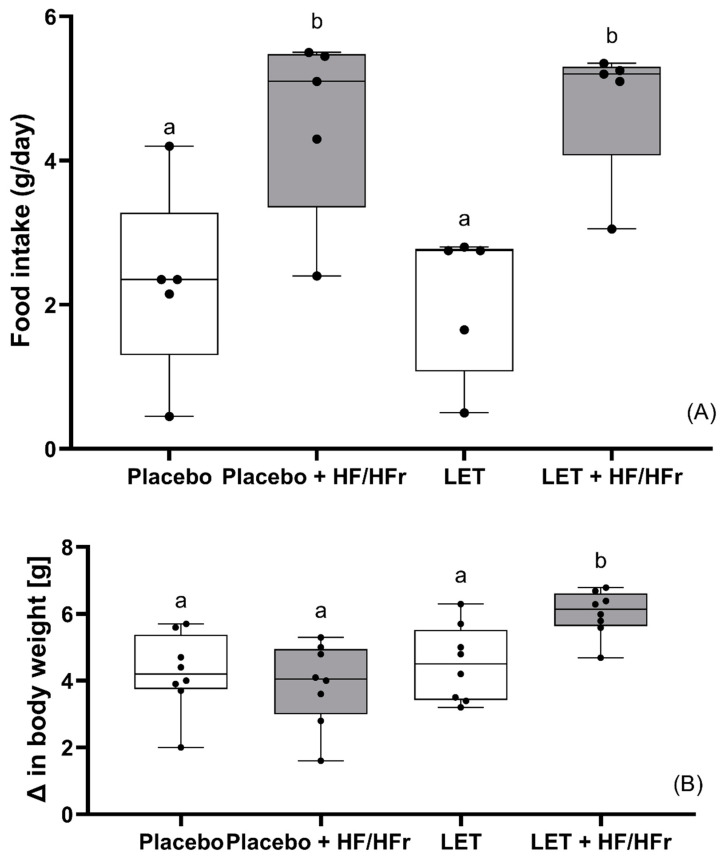
Effects of HF/HFr (high fat/high fructose) diet on food intake (**A**) and changes in body weight (**B**) in placebo or LET-treated (letrozole-treated) mice. The data are presented as medians, with boxes and whiskers representing the interquartile range and the 5th–95th percentiles (*n* = 8 per group), analyzed using one-way ANOVA followed by Tukey’s post hoc test. Values with different letters show statistically significant differences (*p* < 0.05).

**Figure 2 nutrients-14-02478-f002:**
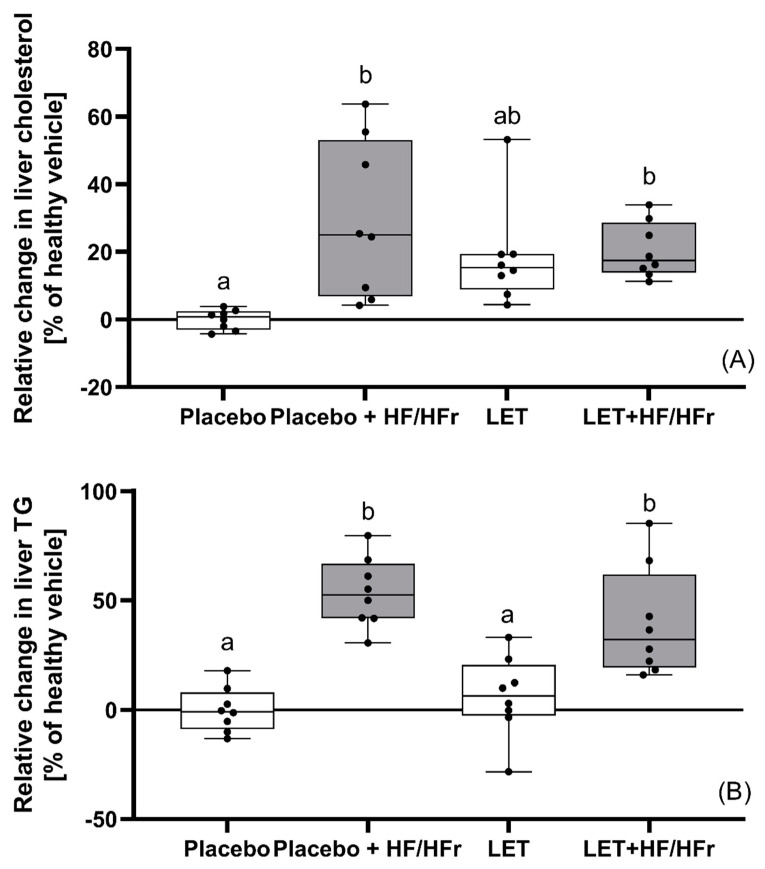
Terminal changes in the liver cholesterol (**A**) and triglycerides (**B**). The data are presented as medians, with boxes and whiskers representing the interquartile range and the 5th−95th percentiles (*n* = 8 per group), analyzed using one−way ANOVA followed by Tukey’s post hoc test. Values with different letters show statistically significant differences (*p* < 0.05).

**Figure 3 nutrients-14-02478-f003:**
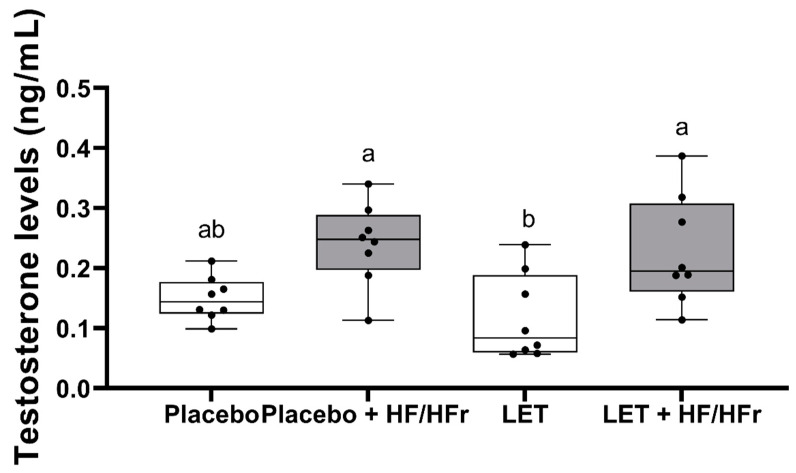
Effects of HF/HFr diet on testosterone levels in placebo or letrozole-treated mice. The data are presented as medians, with boxes and whiskers representing the interquartile range and the 5th−95th percentiles (*n* = 8 per group), analyzed using one-way ANOVA followed by Tukey’s post hoc test. Values with different letters show statistically significant differences (*p* < 0.05).

**Figure 4 nutrients-14-02478-f004:**
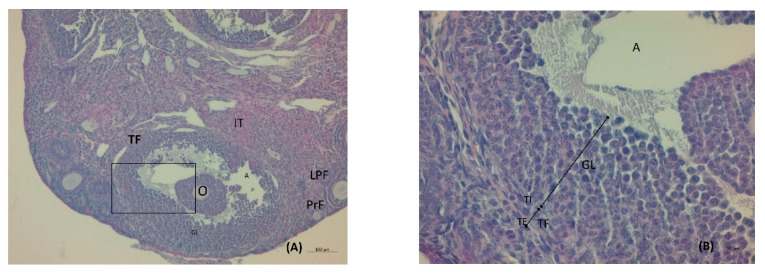
Histological sections of the ovaries of intact mice: tertiary follicle. O: oocyte; GL: granulosa layer; A: antrium; LPF: late primary follicle; PrF: primodial follicle; IT: interstitial tissue; TF: theca folliculi; TE: theca externa; TI: theca interna (H&E; power 100×(**A**)/400×(**B**)).

**Figure 5 nutrients-14-02478-f005:**
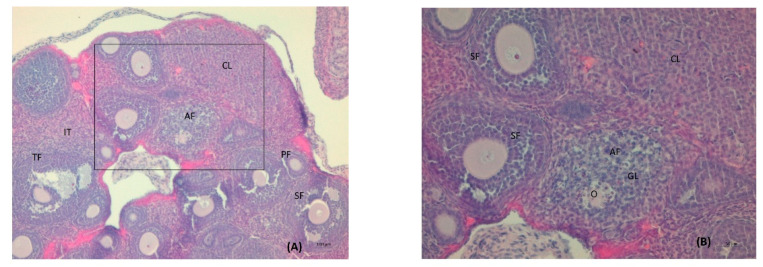
Histological sections of the ovaries with atretic follicles (AF) in intact mice on the HF/HFr diet. TF: tertiary follicle; PF: primary follicle; SF: secondary follicle; CL: corpus luteum; IT: interstitial tissue; O: oocyte; GL: granulosa cells in the process of degradation (H&E; power 100×(**A**)/200×(**B**)).

**Figure 6 nutrients-14-02478-f006:**
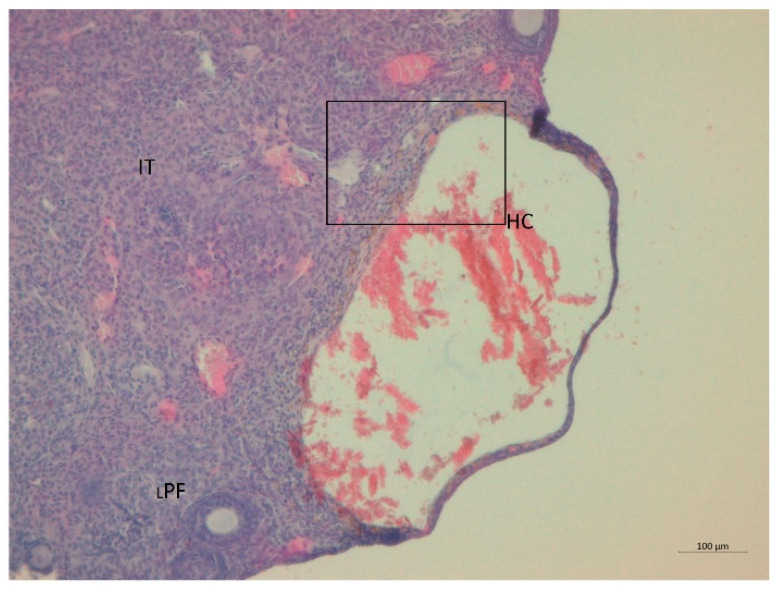
Histological sections of ovaries with hemorrhagic cysts (HC) in LET-treated mice. LPF: late primary follicle; IT: interstitial tissue (H&E; power 100×).

**Figure 7 nutrients-14-02478-f007:**
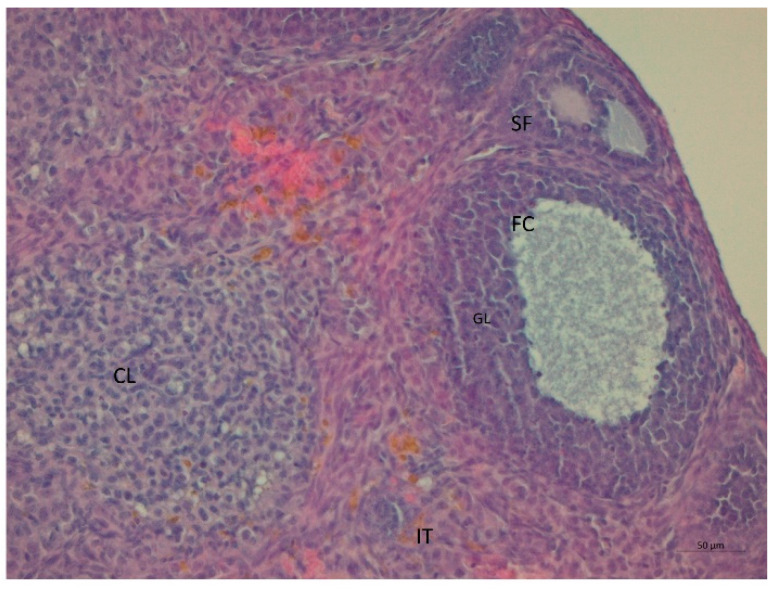
Histological sections of ovaries with follicular cysts (FC) in LET-treated mice. GL: granulosa cells; CL: corpus luteum; IT: interstitial tissue (H&E; power 200×).

**Figure 8 nutrients-14-02478-f008:**
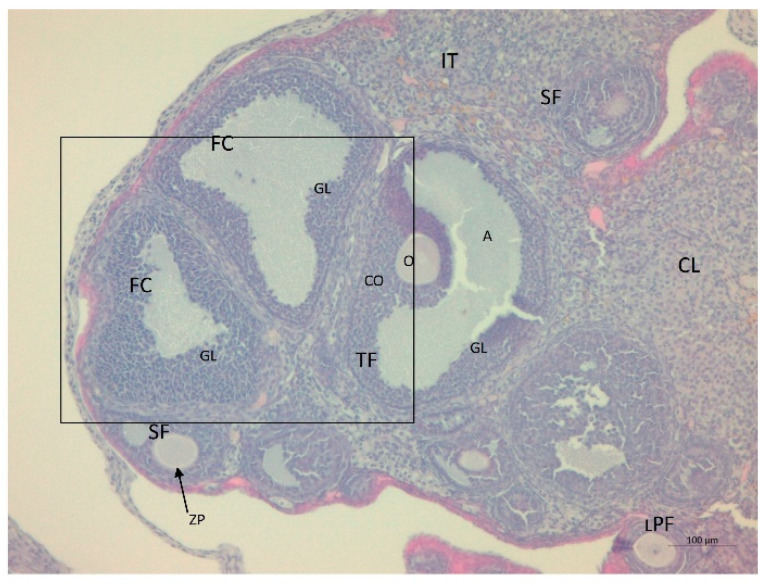
Histological sections of ovaries with follicular cysts (FC) in LET-treated mice on the HF/HFr diet. TF: tertiary follicle; O: oocyte; GL: irregular granulosa layer; CO: cumulus oophorus; A: antrium; LPF: late primary follicle; SF: secondary follicle; ZP: zona pellucida; IT: interstitial tissue (H&E; power 100×).

**Figure 9 nutrients-14-02478-f009:**
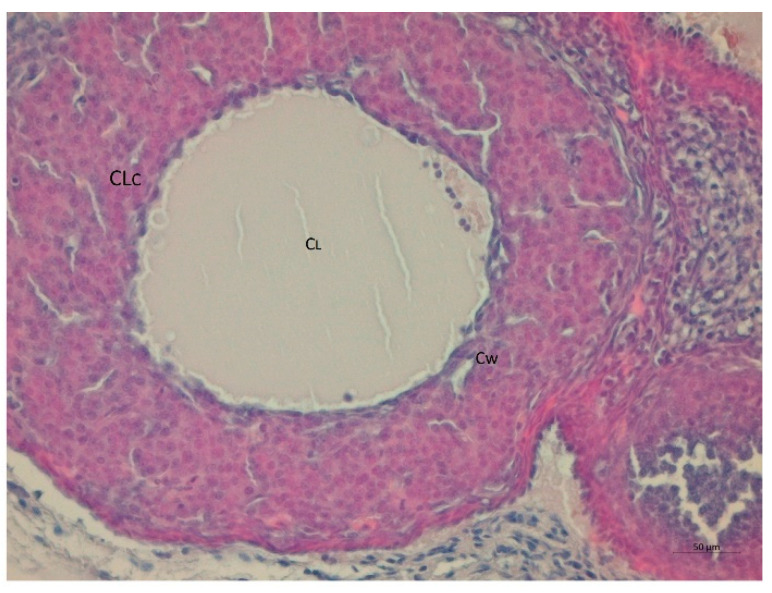
Histological sections of ovaries with corpus luteum cysts (CLc) in LET-treated mice on the HF/HFr diet. Cw: cyst wall; CL: cyst lumen (H&E; power 400×).

**Table 1 nutrients-14-02478-t001:** Effects of the HF/HFr diet on metabolic parameters in placebo and LET-treated mice.

Variables	Placebo		Letrozole	
	Control	HF/HFr	Control	HF/HFr
Total cholesterol (mg/dL)	131.9 ± 21.8 ^a^	188.4 ± 26.0 ^b^	137.6 ± 23.0 ^a^	191.7 ± 39.8 ^b^
LDL cholesterol (mg/dL)	51.6 ± 5.9 ^a^	77.1 ± 4.9 ^b^	49.5 ± 2.5 ^a^	74.5 ± 6.2 ^b^
HDL cholesterol (mg/dL)	30.8 ± 4.1 ^a^	40.0 ± 3.2 ^b^	28.9 ± 1.8 ^a^	38.3 ± 1.6 ^b^
Non-HDL cholesterol (mg/dL)	101.2 ± 22.0 ^a^	148.4 ± 28.3 ^b^	108.7 ± 23.7 ^a^	153.4 ± 40.6 ^b^
Triglycerides (mg/dL)	141.2 ± 16.4	135.9 ± 21.3	136.2 ± 16.6	128.7 ± 12.9
TC/HDL ratio	4.4 ± 0.9	4.8 ± 1.0	4.8 ± 0.9	5.0 ± 1.2
NEFA (mmol/L)	1.09 ± 0.05 ^a^	1.24 ± 0.06 ^b^	1.14 ± 0.11 ^a^	1.12 ± 0.06 ^a^
Glucose (mg/dL)	119.9 ± 14.0 ^ab^	140.1 ± 23.0 ^b^	113.9 ± 12.0 ^a^	136.3 ± 18.4 ^ab^
Insulin (mU/L)	2.9 ± 0.8	3.4 ± 1.5	2.6 ± 0.7	3.4 ± 0.7
HOMA-IR	0.9 ± 0.3 ^ab^	1.0 ± 0.2 ^ab^	0.7 ± 0.2 ^a^	1.2 ± 0.3 ^b^
HOMA-β	18.5 ± 2.9	17.7 ± 10.9	20.3 ± 9.3	20.0 ± 8.0
QUICKI	0.8 ± 0.1 ^ab^	0.7 ± 0.1 ^ab^	0.8 ± 0.1 ^a^	0.7 ± 0.1 ^b^
CRP (mg/L)	25.4 ± 4.2	27.2 ± 1.5	27.5 ± 2.9	29.4 ± 3.0
TAC (μmol/L)	6.5 ± 5.6 ^a^	2.8 ± 1.1 ^ab^	4.9 ± 3.5 ^ab^	1.6 ± 0.5 ^b^

Results are expressed as mean ± SD (*n* = 8 per group). Values with different letters (a,b) show statistically significant differences (*p* < 0.05, Tukey’s post hoc test). NEFA: non-esterified fatty acids; TC: total cholesterol; HDL: high-density lipoprotein; LDL: low-density lipoprotein; HOMA-β: homeostasis model assessment of β-cell function; HOMA-IR: homeostasis model assessment of insulin resistance; QUICKI: quantitative insulin sensitivity check index; CRP: C-reactive protein; TAC: total antioxidant capacity; HF/HFr: high fat, high fructose.

## Data Availability

The datasets used and analyzed in the current study are available from the corresponding author upon reasonable request.

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
