# Peer review of "The Role of a High-Fat, High-Fructose Diet on Letrozole-Induced Polycystic Ovarian Syndrome in Prepubertal Mice"

_nutrients, 2022, doi:10.3390/nu14122478_

Round 1
Reviewer 1 Report
I found this a difficult paper which extrapolates mice findings to humans. The study appears well carried out but I am unclear of the value as it is about a high fat and high fructose diet given to rats and extrapolated to humans an example of a poor diet. Poor diets are high in saturated fat sugar and alcohol, not just fat and fructose. The methodology is good as is the experimentation but not sure that it is valid to draw conclusions about humand from it.
Author Response
We would like to thank the Reviewer for detailed comments and suggestions for the manuscript. We believe that the comments have identified important areas which required improvement. After completion of the suggested edits, the revised manuscript has benefitted from an improvement in the overall presentation and clarity. Below, you will find a point-by-point description of how each comment was addressed in the manuscript.
- I found this a difficult paper that extrapolates mice findings to humans. The study appears well carried out but I am unclear of the value as it is about a high fat and high fructose diet given to rats and extrapolated to humans an example of a poor diet. Poor diets are high in saturated fat sugar and alcohol, not just fat and fructose. The methodology is good as is the experimentation but not sure that it is valid to draw conclusions about human from it.
Answer: Thank you Reviewer for your suggestion. In reference to your comment, we would like to explain why we chose in our experiment a high-fat and high-fructose (HF/HFr) diet. Although definitions of the Western diet vary, this diet is often characterized by a high intake of animal and trans fats, sugar and fructose and a low intake of fiber and phytochemicals (1). As was mentioned in our paper this kind of diet (western diet) is very popular in the modern world, especially among young people (2). Some studies report that the HF/HFr diet reflects the macronutrient composition of a Western diet (3). Moreover, in the animal model, fructose was the best choice among simple sugars because of its most potent metabolic disturbance-inducing effect in comparison to e.g. glucose (4). Of course, we can also find studies where the Western dietary pattern is associated also with higher alcohol consumption (5) but due to the awareness that alcohol is prohibited among children, we did not decide to investigate the effect of the poor diet including alcohol in this animal experiment.
(1) Oddy, Wendy H, Carly E Herbison, Peter Jacoby, Gina L Ambrosini, Therese A O’Sullivan, Oyekoya T Ayonrinde, John K Olynyk, i in. 2013. „The Western Dietary Pattern Is Prospectively Associated With Nonalcoholic Fatty Liver Disease in Adolescence”. American Journal of Gastroenterology 108 (5): 778–85. https://doi.org/10.1038/ajg.2013.95.
(2) Bibiloni, Maria del Mar, Jordi Pich, Antoni Pons, i Josep A Tur. 2013. „Body Image and Eating Patterns among Adolescents”. BMC Public Health 13 (1): 1104. https://doi.org/10.1186/1471-2458-13-1104
(3) Coate, Katie Colbert, Guillaume Kraft, Margaret Lautz, Marta Smith, Doss W. Neal, i Alan D. Cherrington. 2011. „A High-Fat, High-Fructose Diet Accelerates Nutrient Absorption and Impairs Net Hepatic Glucose Uptake in Response to a Mixed Meal in Partially Pancreatectomized Dogs”. The Journal of Nutrition 141 (9): 1643–51. https://doi.org/10.3945/jn.111.145359.
(4)Wong, Sok Kuan, Kok-Yong Chin, Farihah Hj Suhaimi, Ahmad Fairus, i Soelaiman Ima-Nirwana. 2016. „Animal Models of Metabolic Syndrome: A Review”. Nutrition & Metabolism 13 (1): 65. https://doi.org/10.1186/s12986-016-0123-9
(5) Sasaki, Mariko, Naoko Miyagawa, Sei Harada, Kazuo Tsubota, Toru Takebayashi, Yuji Nishiwaki, i Ryo Kawasaki. 2022. „Dietary Patterns and Their Associations with Intermediate Age-Related Macular Degeneration in a Japanese Population”. Journal of Clinical Medicine 11 (6): 1617. https://doi.org/10.3390/jcm11061617.
Reviewer 2 Report
The study is interesting showing new insights related to PCOS and in the liver. It is interesting that authors chose the liver considering the toxicity.
1. In the description of groups (Lines 98-102), authors must let clear that the subcutaneous implant of placebo or letrozole is to induce the PCOS. In the way written in the text it looks like additional groups are performed with the implant.
2. The description of symbols related to differences should be shown in each legend and not in the text of the results (lines 176-182). Also, it is too much symbols that let the text more confuse, mainly in the figure 1. Moreover, the comparison are always performed comparing the treated with control (placebo) so the symbol should be on the “treated” group and not on the control group. For example the “a” on intact group in figure 1.
3. Why not include the figures provided in the supplementary material in the main file of the manuscript?
Minor points
1. In the introduction authors must explain the abbreviation LET when cited for the first time. In the text is just cited in the abstract and not in the introduction.
2. Include in the methodology that the present study was approved by a ethical committee.
3. In the line 113 is “three 3 days”. Remove one of the 3s or use (3) in parenthesis.
4. Add a final dot “.” after “…Hungary)” in line 126.
5. For future studies authors may consider to analyze hepatic enzymes too.
Author Response
We would like to thank the Reviewer for detailed comments and suggestions for the manuscript. We believe that the comments have identified important areas which required improvement. After completion of the suggested edits, the revised manuscript has benefitted from an improvement in the overall presentation and clarity. Below, you will find a point-by-point description of how each comment was addressed in the manuscript.
The study is interesting showing new insights related to PCOS and in the liver. It is interesting that the authors chose the liver considering the toxicity.
- In the description of groups (Lines 98-102), the authors must let clear that the subcutaneous implant of placebo or letrozole is to induce the PCOS. In the way written in the text, it looks like additional groups are performed with the implant.
After acclimatization, at four weeks of age, the mice were randomly assigned to four groups: (1) mice receiving placebo pellet fed a standard chow diet (n = 8); (2) mice receiving placebo pellet fed the HF/HFr diet (n = 8); (3) mice receiving LET pellet fed a standard chow diet (n = 8); and (4) mice receiving LET pellet fed the HF/HFr diet (n = 8). Subcutaneous implantation of continuous release letrozole (3 mg, 50 μg / day) or placebo pellet was performed to induce PCOS or form a control group, respectively.
- The description of symbols related to differences should be shown in each legend and not in the text of the results (lines 176-182). Also, it is too much symbols that let the text more confuse, mainly in the figure 1. Moreover, the comparison are always performed comparing the treated with control (placebo) so the symbol should be on the “treated” group and not on the control group. For example the “a” on intact group in figure 1.
Answer: Thank you Reviewer for your suggestion. According to the Reviewer`s suggestion description of symbols in the Table and Figures has been changed to make the article more clearer to “Values with different letters show statistically significant differences (p < 0.05).”Additionally, we replaced the word „intact” with „placebo” to make the article clearer. The description of the symbols has been moved to the legend of each plot.
- Why not include the figures provided in the supplementary material in the main file of the manuscript?
Answer: Thank you Reviewer for your suggestion. According to the Reviewer`s suggestion the Figures were transferred from the Supplementary Material to the main text of the manuscript.
Minor points
- In the introduction authors must explain the abbreviation LET when cited for the first time. In the text is just cited in the abstract and not in the introduction.
Answer: Thank you for your suggestion. According to the Reviewer, the abbreviation LET has been explained in the main text.
- Include in the methodology that the present study was approved by a ethical committee.
Answer: Thank you for your suggestion. According to the Reviewer`s information that the present study was approved by an ethical committee has been added to the methodology section.
- In the line 113 is “three 3 days”. Remove one of the 3s or use (3) in parenthesis.
Answer: Thank you Reviewer for your suggestion. The word “three” has been removed and the word “for” has been added.
- Add a final dot “.” after “…Hungary)” in line 126.
Answer: Thank you for your suggestion. The dot has been added.
- For future study’s authors may consider to analyze hepatic enzymes too.
Answer: Thank you for Reviewer suggestion. We agree with Reviewer concept that analyzing hepatic enzymes is important to precisely conclusions, but the amount of serum obtained from mice allows us for a limited number of analyzes focused only on parameters strictly associated with PCOS.
Reviewer 3 Report
The study investigates the effect of a high-fat and high-fructose diet on hormonal and metabolic dysregulation in PCOs.
The study was well planned and performed and presented results are very interesting for clinical practice. The findings suggest that unappropriate diet mat have a great impact on PCOs progression or even development.
Author Response
We would like to thank Reviewer for reviewing and appreciating our study.
Round 2
Reviewer 1 Report
I find it difficult to equate a poor diet with high fructose which is usually from fruit plus part of sucrose --can you add a sentence about fructose and where it is found
Author Response
We would like to thank the Reviewer for detailed comments and suggestions for the manuscript. We believe that the comments have identified important areas which required improvement. After completion of the suggested edits, the revised manuscript has benefitted from an improvement in the overall presentation and clarity. Below, you will find a description of how each comment was addressed in the manuscript.
- I find it difficult to equate a poor diet with high fructose which is usually from fruit plus part of sucrose -can you add a sentence about fructose and where it is found.
Answer: Thank you Reviewer for suggestion. According to the Reviewer's suggestion, the following sentences have been added.
At present, the young people are increasingly subjected to significant exposure to diet rich in saturated fat, sugar, and fructose in particular (1). Soft drinks, energy drinks, fruit juices and nectars, and generally free sugars in liquid form represent the main source of fructose consumed by today’s young generation, especially obese one. Of course, fructose is contained also in fruit, but in minimal quantities compared to the weight of the fruit itself. (2)
- Munt AE, Partridge SR, Allman-Farinelli M. The barriers and enablers of healthy eating among young adults: a missing piece of the obesity puzzle: A scoping review: Barriers and enablers of healthy eating. Obes Rev. styczeń 2017;18(1):1–17.
- Giussani M, Lieti G, Orlando A, Parati G, Genovesi S. Fructose Intake, Hypertension and Cardiometabolic Risk Factors in Children and Adolescents: From Pathophysiology to Clinical Aspects. A Narrative Review. Front Med. 12 kwiecień 2022;9:792949.